# Educating Parents Improves Their Ability to Recognize Adolescent Idiopathic Scoliosis: A Diagnostic Accuracy Study

**DOI:** 10.3390/children9040563

**Published:** 2022-04-15

**Authors:** Charlotte de Groot, Johan L. Heemskerk, Nienke W. Willigenburg, Mark C. Altena, Diederik H. R. Kempen

**Affiliations:** 1Department of Orthopedic and Trauma Surgery, OLVG, P.O. Box 95500, 1090 HM Amsterdam, The Netherlands; j.l.heemskerk@olvg.nl (J.L.H.); n.w.willigenburg@olvg.nl (N.W.W.); m.c.altena@olvg.nl (M.C.A.); d.h.r.kempen@olvg.nl (D.H.R.K.); 2Department of Orthopedic Surgery, Amsterdam University Medical Center, 1105 AZ Amsterdam, The Netherlands

**Keywords:** adolescent idiopathic scoliosis, Adam’s forward bend test, scoliosis screening, untrained parents, self-detection, self-screening, education, informational support, sensitivity, specificity

## Abstract

(1) In countries where scoliosis screening programs ended, the responsibility for detection shifted from healthcare professionals to parents. Since recognizing scoliosis is difficult for parents, more patients are presenting late. Increased awareness of scoliosis may favor earlier detection. This study examines the effect of educating parents to recognize scoliosis. (2) In this cross-sectional study a consecutive group of parents completed a digital assessment. They had to complete two identical series of fourteen cases (eight with scoliosis and six without). Each case displayed two photographs of the child’s back; one in standing position and one during forward-bending. Based on visual inspection, parents had to indicate if the child had to be referred to a physician. After assessing the first series, information was given on how to detect scoliosis. Subsequently, parents assessed the second series of cases. Sensitivity and specificity were calculated before and after education. (3) A total of 100 parents completed the assessment. The sensitivity to detect scoliosis was slightly but significantly higher after education (68.8% versus 74.0%; *p* = 0.002), while specificity was not (74.0% versus 74.8%; *p* = 0.457). (4) This study showed that educating parents improved their ability to recognize scoliosis without increasing the false positive referral rate. Although written instructions can bridge the gap with professional screening programs, the overall sensitivity in this study remained low. Therefore, education can improve the awareness and ability to detect scoliosis, but will not replace screening by professionals.

## 1. Introduction

Screening for adolescent idiopathic scoliosis (AIS) is based on the assumption that early detection is beneficial. While small curves are common and of limited clinical relevance, moderate curvatures warrant treatment to prevent progression during skeletal growth [1]. Bracing is the only treatment proven in preventing curve progression and aims to keep the curve below 45–50 degrees at skeletal maturity [2]. Larger curves have a high risk of progression during adulthood and are associated with adverse outcomes (e.g., pulmonary disorders, decreased self-image, back pain) [3,4]. Therefore, patients with curves above 50 degrees are generally treated surgically. Strong evidence exists that early brace treatment of moderate scoliosis decreases the risk of curve progression and thereby prevents curvatures of large magnitude [2,5].

Despite the rationale for preventing large curvatures, some countries ended their professional scoliosis screening [6,7,8,9]. In the absence of professional scoliosis screening, scoliosis has to be detected by untrained family or friends, and the proportion of patients presenting in an advanced stage to the orthopedic specialist has increased significantly [10,11,12,13,14]. Non-operative treatments are less effective in more advanced curvatures (>40 degrees), resulting in an increased frequency of patients who undergo corrective surgery [11,12]. This observation highlights the importance of early scoliosis detection [15].

The shifted responsibility of scoliosis detection from professional to untrained adult impacts the appropriateness of AIS referral [11,12,13,14]. Delayed referral to a physician may be due to a lack of knowledge and awareness among untrained adults. Whether educating parents increases their ability to recognize scoliosis has never been investigated. Therefore, this study compares the ability of parents to recognize scoliosis on pictures of children in the upright and bending position, before and after providing information on the outward signs of scoliosis. We hypothesized that educational information about scoliosis improves the accuracy of scoliosis detection.

## 2. Materials and Methods

This cross-sectional study was performed following the STARD checklist (Standards for Reporting of Diagnostic Accuracy Studies) to ensure proper reporting and improve the methodological quality [16]. Survey assessors had to fill out a digital diagnostic assessment containing photographs of children with and without scoliosis. The protocol was registered in the (blinded) register for clinical trials (blinded) and was approved by the local Medical Ethics Committee (WO 16.017) before initiation. All survey assessors provided informed consent before participation in the study. Informed consent was also granted by parents and children for using their photographs in the survey.

### 2.1. Assessors of the Survey

Assessors enrolled in this study were consecutively recruited from the pediatric orthopedic outpatient clinic (hip dysplasia) in our hospital. All parents visiting the outpatient clinic with their newborns were eligible. Parents visiting the scoliosis outpatient clinic were not eligible to participate in the study because of their familiarity with this deformity. The sample size was based on feasibility considerations, and we assumed that a sample of >100 assessors would suffice to detect any substantial difference before and after the informational support. No specific power analyses were possible since well-supported assumptions to use as input for such an accurate sample size calculation were lacking. After providing informed consent, the assessors completed the diagnostic assessment (details below) on a desktop computer.

### 2.2. Digital Diagnostic Assessment

A digital diagnostic assessment was created using QuestManager software (Vital Health). The assessment contained 2 series of 14 slides of photographs (28 slides for the total assessment). Each slide had two photographs of the same child. One photograph was of the back of the child in an upright position, and the other was in Adam’s forward-bending position (Figure 1). All photographs were taken by a medical photographer experienced in the imaging of scoliosis. For the Adam’s bending photographs, the amount of flexion varied for thoracic and lumbar curves to optimize the visualization of the gibbus. The photographs were cropped just below the child’s hairline to protect their privacy. On each slide, the assessors had to answer the following question: “Does this child have an abnormality for which you would consider consulting a physician?”. If this question was answered with a yes, the following question showed up: “Can you highlight the most outstanding abnormality on the photograph that prompted consultation of a physician?”. After the first series of photographs, a slide with information about the signs of scoliosis was shown to educate the parents (Figure 2). The education sheet has not been validated as an educational tool. After reading the information, assessors completed the second part. The second series of photographs were identical to the first part of the survey.

Two survey versions were constructed, each with a different set of children, to decrease the influence of individual cases on the detection rate. In both surveys fourteen children were shown including six children with no spinal deformity, six patients with a thoracic scoliosis (Cobb angle of 10–20° (*n* = 2), 20–30° (*n* = 2), 30–40° (*n* = 2)) and two patients with a lumbar scoliosis (20–30° (*n* = 1), 30–40° (*n* = 1)). The absence of a deformity in the non-scoliotic patients was confirmed on X-rays taken during trauma screening or for the exclusion of spinal abnormalities in patients with back pain. Characteristics of the children represented in the photographs are reported in the Appendix A.

### 2.3. Target Condition, Index Test, and Reference Standard

The target condition was AIS. AIS develops in children between 10 and 18 years, and it is defined as a curvature of the spine of more than 10° in the coronal plane. The index test was a visual inspection of two photographs of a child’s trunk. The reference standard (not available to the assessors) was a coronal posterior-anterior x-ray of the whole spine, which is currently the gold standard in quantifying the magnitude of scoliosis curves. All x-rays were evaluated by a radiologist not involved in this study. The clinical photos were taken on the same day as the full spine radiograph.

### 2.4. Outcome

The screening accuracy was calculated before and after providing the information and was compared with the gold standard (full spine radiograph, obtained within standard care). The accuracy of scoliosis detection was quantified by sensitivity, specificity, likelihood ratios (negative and positive), and diagnostic odds ratio (DOR). These characteristics are intrinsic to the test and are independent of prevalence.

The secondary aim of the study was to perform a sub-analysis in which the scoliosis detections were evaluated by curve type and curve severity. Additionally, we explored whether education induced a shift of the most highlighted external characteristic before and after providing information about the deformity. This outcome was visualized by plotting the highlighted areas over a picture and was quantified by counting the number of highlighted characteristics for all AIS patients.

### 2.5. Statistical Analysis

Data was collected using QuestManager (Vital Health), and statistical analyses were performed using SAS enterprise guide, version 7.15 (SAS Institute Inc, Cary, NC, USA), and MedCalc, version 19.6.3 (MedCalc Software Ltd., Ostend, Belgium). Continuous normally distributed data were reported as mean and standard deviation (SD), and categorical data were reported as frequencies and percentages. The screening results were plotted against the reference standard (full spine radiography) in cross tables. All estimates were accompanied by an exact 95% confidence interval based on binomial distributions. Differences in sensitivity and specificity (before-and-after measurements) were tested with the McNemar test in the case of two categories or the McNemar–Bowker test in the case of several categories. Significance was set at α < 0.05 for all analyses.

## 3. Results

### 3.1. Characteristics of Assessors

One hundred participants completed the survey. The mean age (SD) of the assessors was 36.2 (7.8) years, and 69% were female. The majority had an educational level lower than a master’s degree (56%), did not work in healthcare (81%), and had one or two children (87%) (Table 1).

### 3.2. Accuracy of Scoliosis Detection

The detection rate of scoliosis (abnormality for which consultation of a physician was considered) of the assessors before educating them about scoliosis was 50.8% (Table 2). The corresponding sensitivity was 68.8% (95% CI, 65.4–72.0) and the specificity was 74.0% (95% CI, 70.8–77.0) (Table 3). After educating them on how to detect scoliosis, the detection of abnormalities increased to 53.1%, the corresponding sensitivity was 74.0 (95% CI, 70.8–77.0), and the specificity was 74.8% (95% CI, 71.2–78.3). The sensitivity increased significantly with 5.25% (95% CI, 2.1–8.42; *p* = 0.002), while the specificity did not change significantly (95% CI, −2.3–5.61; *p* = 0.457). The false positive detection rate was 26.8% (95% CI, 23.3–30.6%) before, and 25.2% (95% CI, 21.7–28.8%; *p* = 0.457) after educating the assessors about the signs of scoliosis.

### 3.3. Scoliosis Recognition Stratified by Curve Type and Severity

Children with thoracic scoliosis were more often recognized by parents compared to patients with lumbar scoliosis. Children with thoracic scoliosis were correctly detected in 73.5% of the cases before providing information about scoliosis, and in 76.3% after education. (*p* = 0.117). The effect of education was stronger for patients with lumbar scoliosis, in which the rate of scoliosis detection increased from 54.5% to 67.0% (*p* < 0.001). An exploratory sub-analysis was performed to determine the scoliosis detection rate for various curve severities (i.e., a stepwise increment of 10 degrees). After educating assessors, the detection rate increased for all categories except for patients with a thoracic curve between 10 and 20 degrees (Figure 3).

### 3.4. Scoliosis Characteristics

When the assessors recognized an abnormality, they were asked to highlight the most outstanding visual characteristic. Figure 4 visualizes the highlighted areas of the assessors for one of the patients in the diagnostic assessment. The results were quantified by counting the dots in each region (Table 4) for all AIS patients. The most highlighted regions before education were the gibbus (region 5: 195 times highlighted, 24.4%), followed by the waist (region 4: 115 times highlighted, 14.4%) and spine (region 3: 111 times highlighted, 13.9%). After education, the most highlighted regions were still the gibbus (region 5: 201 times highlighted, 25.1%), the waist (region 4: 151 times highlighted, 18.9%), and the spine (region 3: 106 times highlighted, 13.3%). The observed frequencies were not significantly different before and after education (*p* = 0.707). 

## 4. Discussion

This study revealed that parents’ ability to detect scoliosis improved after a brief education about the signs of the deformity. The true positive rate (sensitivity) increased slightly but significantly from 68.8% to 74.0% without increasing the false positive rate (26.8% to 25.2%). These results suggest that adequate referral of patients with scoliosis to an orthopedic surgeon may be facilitated by increasing knowledge and awareness about the outward signs of scoliosis among parents.

Due to the shifted responsibility of scoliosis detection, the characteristics of AIS patients in the orthopedic clinic have changed [10,11,12,13,14]. More patients visit the clinic in a later stage with larger curvatures. Screening by parents could be a potential strategy to improve scoliosis detection in healthcare systems without professional screening. Melanoma care is an example of another healthcare system where short cognitive education with photographs effectively improves the ability of laypersons to recognize melanoma among benign lesions and proved to be one of the most cost-effective methods to improve melanoma prognosis [17]. The idea of screening by parents is not new and is already implemented in the ‘National Self-Detection Program’ of the Spine Society of Australia [18]. This program is based on the distribution of a ‘Self-Detection Fact Sheet’ in schools to educate children and their parents about the signs of scoliosis [19]. Nevertheless, AIS is still detected relatively late in Australia, despite the implementation of the national self-detection screening program [13]. Based on our results, the education of parents seems beneficial in improving scoliosis detection. The limited effect of education in this study is not surprising since a previous study by our team showed a difference of 10% in sensitivity using a similar study setup between untrained parents and healthcare professionals [20]. Education of parents can thus partially decrease the gap in sensitivity of recognizing the deformity compared to healthcare professionals.

The potential benefit of earlier detection is supported by the SRS, AAOS, POSNA, and AAP since it allows non-surgical (i.e., brace) treatment to prevent progression [15]. However, several national committees (e.g., the US Preventive Task Force (USPTF), the UK National Screening Committee (UK NSC), and the Dutch TNO) have stated that the current screening programs were insufficient. One of their arguments is the limited sensitivity and positive predictive value (PPV) of the Adams forward-bending test [6,7,8,20,21]. Although the education of parents can bridge the gap with professional screening programs, the overall sensitivity of the test remained low in this study. However, false positive rates did not change and remained substantial (26.8% vs. 25.2%). Therefore, it cannot be expected that the education of parents will solve the concerns of opponents of screening about the false positive referrals and possible overtreatment of children [6,7,8,21,22].

The negative consequences of the discontinuation of professional screening are becoming evident, and a situation with no screening regulation seems suboptimal. More AIS patients are presenting with large curvatures at the orthopedic clinic, resulting in more patients requiring surgical correction. This situation is not only more invasive for the patient, but it also increases hospital expenses [23]. Strong evidence has become available about the effectiveness of bracing in preventing scoliosis progression [2]. Bracing achieves better results with smaller curvatures, advocating the need for early detection [15]. With these new insights, the discussion about AIS screening should be focused on a different approach to improve the early detection of scoliosis.

The main limitation of our study is that the visual inspection of the trunk was based on two photographs instead of a real-life clinical assessment. The outward signs of scoliosis may be more evident during a 3D inspection, and a bending test may be more informative when it is dynamic. Unfortunately, real-life screening was not possible since it was not possible to bring all children and parents together at the same time and location for the screening. Although a medical photographer took the pictures and tried to maximally visualize the hump, the use of static photographs probably limits the sensitivity and specificity of scoliosis detection. Therefore, the effect of training may be more pronounced in a real-life 3D setting. Massive educational programs in real-life may further bridge the gap between trained professionals and untrained parents. Furthermore, patients were stratified by curvature magnitude, and the curve magnitude does not necessarily correlate with the external characteristics of scoliosis. Scoliosis screening is a subjective assessment based on the outward signs of scoliosis. Sometimes outward signs can be more prominent in patients with smaller curvatures compared to patients with larger curvatures. It is possible that other photographs with different patients with the same curve magnitude would result in different screening measures because of the variation in prominence of the outwards signs of scoliosis. More research is necessary to investigate the merits of scoliosis screening by parents, whether the accuracy measures reported in this study can be generalized to real-life situations, and whether this eventually results in less surgery and better patient outcomes. Furthermore, future studies are needed to investigate which type of instruction educates parents best.

## 5. Conclusions

This study showed that the education of parents improved their ability to detect scoliosis from 68.8% to 74.0%, without increasing the false positive referral rates (26.8% before and 25.2% after education). Although written instructions can bridge the gap with professional screening programs, the overall sensitivity in this study remained low. Therefore, education can improve the awareness and ability to detect scoliosis, but will not replace screening by professionals. Education of the parents could be a strategy to facilitate adequate referral of patients with scoliosis to a physician in healthcare systems without professional screening.

## Figures and Tables

**Figure 1 children-09-00563-f001:**
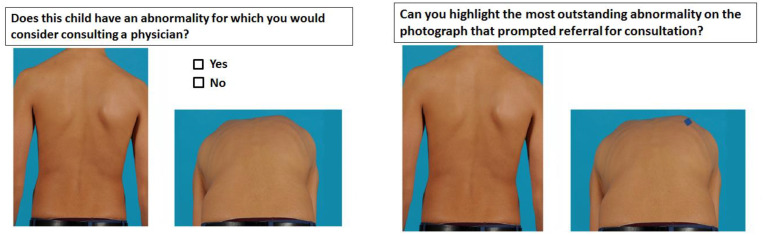
Format diagnostic assessment.

**Figure 2 children-09-00563-f002:**
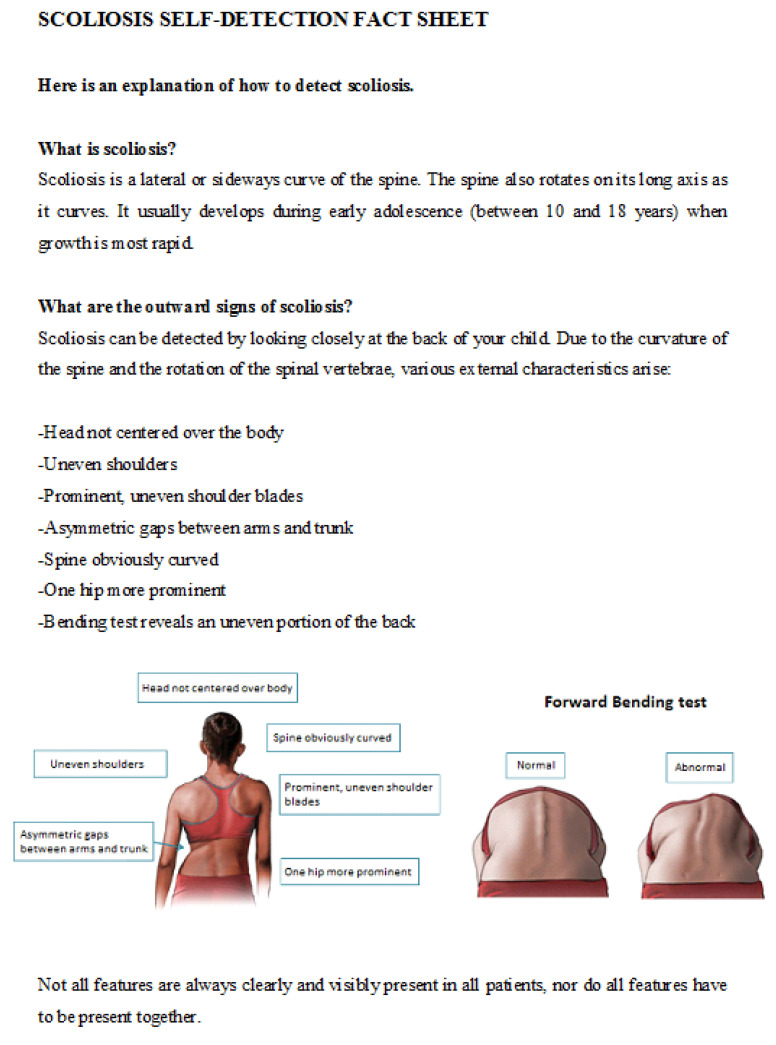
Educational information about scoliosis.

**Figure 3 children-09-00563-f003:**
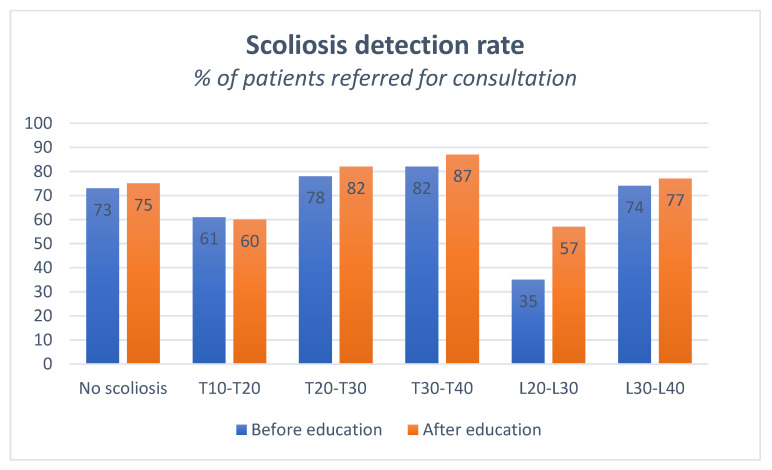
Scoliosis detection rate by severity and curve type.

**Figure 4 children-09-00563-f004:**
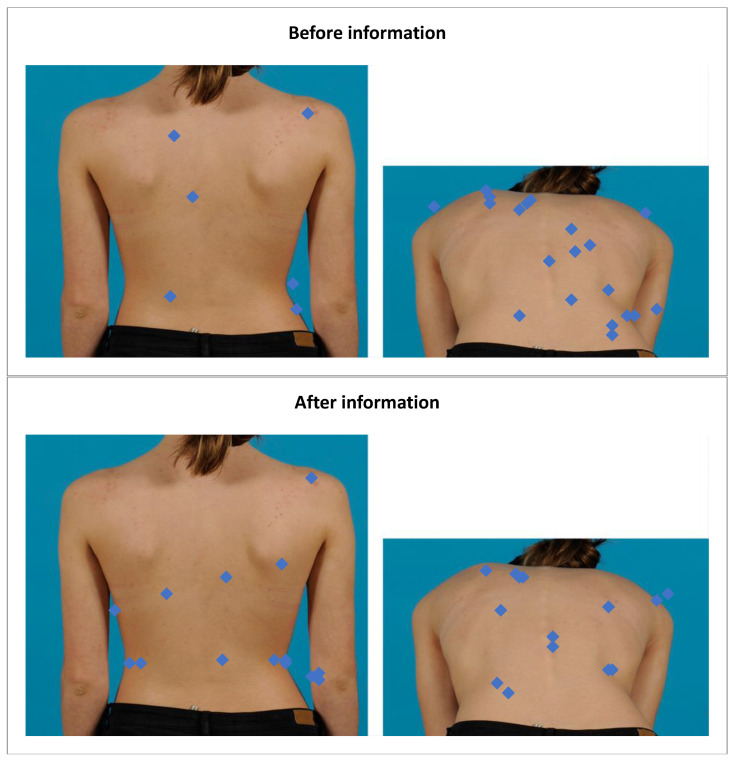
Most outstanding outward sign to consult a physician.

**Table 1 children-09-00563-t001:** Characteristics of survey participants.

Characteristics	*n* = 100
Mean Age, y (sd)	36.2 (7.8)
Female sex, *n* (%)	69
Education level, *n* (%)	
Elementary school	2
High school	22
Vocational education	4
Bachelor degree	28
Master degree or higher	44
Working in healthcare, *n* (%)	19
Number of children, *n* (%)	
1	46
2	41
3	10
4	3

**Table 2 children-09-00563-t002:** Results of scoliosis screening by parents.

**a. Before Education about the Signs of Scoliosis**			
**Ought to be Referred for Consulting a Physician**	**Scoliosis on X-ray**		
	**Present**	**Absent**	**Total**
Yes	550	161	711
No	250	439	689
Total	800	600	1400
**b. After Education about the Signs of Scoliosis**			
**Ought to be Referred for Consulting a Physician**	**Scoliosis on X-ray**		
	**Present**	**Absent**	**Total**
Yes	592	151	743
No	208	449	657
Total	800	600	1400

**Table 3 children-09-00563-t003:** Accuracy of scoliosis detection.

	Before Education	After Education
Sensitivity (%)	68.8 (65.4–72.0)	74.0 (70.8–77.0)
Specificity (%)	73.2 (69.4–76.7)	74.8 (71.2–78.3)
Positive Likelihood Ratio	2.56 (2.23–2.95)	2.94 (2.55–3.40)
Negative Likelihood Ratio	0.43 (0.38–0.48)	0.35 (0.31–0.39)
Diagnostic Odds Ratio	6.00 (4.75–7.58)	8.46 (6.64–10.79)
Values between brackets are 95% confidence interval		

**Table 4 children-09-00563-t004:** Percentages of scoliosis characteristics before and after education.

		Before Education		After Education		*p*-Value ^1^
Anatomic Location	Area Code	Most Reported Outwards Sign	Percentage	Most Reported Outwards Sign	Percentage	
Shoulders	1	45	5.6%	57	7.1%	
Scapulae	2	66	8.3%	60	7.5%	
Spine	3	111	13.9%	106	13.3%	
Waist	4	115	14.4%	151	18.9%	
Gibbus	5	195	24.4%	201	25.1%	
Outside region	6	18	2.3%	17	2.1%	
Not referred for physican consultation	7	250	31.3%	208	26.0%	
Total time highlighted		550		592		
Total area		800	100%	800	100%	0.707

^1^ McNemar-Bowker Test.

## Data Availability

The data and materials are available to strengthen the transparency and reliability of the study. The additional data can be requested by emailing the corresponding author.

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
