# Peer review of "Educating Parents Improves Their Ability to Recognize Adolescent Idiopathic Scoliosis: A Diagnostic Accuracy Study"

_children, 2022, doi:10.3390/children9040563_

Round 1

Reviewer 1 Report

Dear Authors

I like the goal of this paper very much. Early detection of scoliosis to allow early treatment is a laudable goal. I have several question and comments 

  1. line 82- The supplemental data lists 16 AIS patients / 12 controls of 28 patients not 20 AIS patients. Please correct this error and re state your power analysis . Sample size based on "feasibility" Line 76 probably should not use the term "power" in line 81. 
  2. line 88-91- example patient photographs in the Adam's forward flexion position are in slight flexion to show thoracic asymmetry but would need further flexion at waist to show lumber asymmetry. Were photos done to show lumbar spine? 
  3. line 96 and Figure 2.  The education sheet is very nice and clear to me. However, have you validated this education tool as being adequate for instruction to lay persons to distinguish scoliosis from normal? Did you do a pilot on the quality of the instruction sheet? Were the parents offered verbal instruction or did they just read it on their own? 
  4. Line 106- 112. Spine radiographs were only done on AIS patients. How did you declare that non AIS patients indeed did not have mild scoliosis in the absence of an x-ray? 
  5. line 164 Scoliosis characteristics - It is unusual that parents did not mention should asymmetry as this is the most common observation noted by parents in my practice. This may be an unexpected result of cropping the photos at the neck line. Why did you believe that cropping out the posterior head was necessary to protect privacy? 
  6. Line 192 - 197 Did the results cited in discussion meet the a priori parameters on line 80-82?
  7. line 208-211- Your mention reference 23 would seem to be supported by this cohort. However, another conclusion would be that the educational tool was not sufficient to improve the ability of parents to recognize torso deformity and that a better educational tool should be developed and studied . Examples might be a video, on- line instruction , or etc. In addition, use of an inclinometer from a smart phone has be advocated . 
  8. line 215- In 2018, the USPTF did not recommend against scoliosis screening as it stated that evidence was " insufficient ", please correct this statement. 
  9. line 251- 257 Conclusion should have stronger statement that parental education based on a written instruction sheet used in his study had the ability to slightly improve detection of lumbar scoliosis but it was insufficient to reduce the false negative rate. Educational materials to parent should be improved if healthcare system is to rely on parental screening of scoliosis. A more appropriate screening program via primary care physician has been advocated by multiple professional organizations that care for children as mentioned on line 213-14. 
  10. The abstract should be updated to reflect the revised conclusion

Reviewer 2 Report

The article examines the potential benefits of parental traning in the detection of adolescent scoliosis. This training could represent an alternative screening method in countries where screening by physicians is not possible.
The topic is certainly interesting. The article is well written. The methods are comprehensive. The results are clear. The conclusions are consistent with the results.
However, I believe that some aspects useful to readers should be further discussed.
1) Why was it not considered that video rather than static images could have been presented to study participants?
2) the authors report that in Australia screening by trained parents was ineffective in anticipating the time of diagnosis. What could be the reasons for this?
3) it would be helpful if the authors reported other examples of healthcare systems where there is screening by healthcare professionals or screening by educated parents (if any).
4) I think it would be appropriate to point out the most obvious limitation of this study: the setting in which parental education about the characteristics of scoliosis is produced in the study differ substantially from those possible outside a study. I think that the difference in the ability to detect scoliosis between trained and untrained parents is unquestionable, but relatively small although examined in an ideal context. It would be helpful if the authors would further discuss the issues related to the possibilities of effectively educating parents extensively and highlight that the results of massive education could be very different from those found in the study. Nevertheless, I believe that the idea underlying this study is extremely valid and the results worthy of recognition if only for the possibility of increasing readers' awareness of the topic.

Reviewer 3 Report

Thank you for the invitation to review the manuscript titled: EDUCATING PARENTS IMPROVES THEIR ABILITY TO RECOGNIZE ADOLESCENT IDIOPATHIC SCOLIOSIS: A DIAGNOSTIC ACCURACY STUDY [ORIGINAL RESEARCH]

The study was well done.

TITLE:

Main idea is clear and concise

ABSTRACT

- Coherent and readable

- Structured format

INTRODUCTION

- Previous pertinent literature cited and discussed

- Purpose/research hypotheses clearly stated

- Conceptualization and rationale of study clearly apparent

MATERIALs AND METHODS

- Study design appropriate to achieve study objective

- Study population clearly and adequately described

- Sampling procedures appropriate and sufficiently described

- Statistical analyses appropriate and used appropriately

- Statistical analysis accurate

RESULTS

Test statistics adequate

DISCUSSION

- Support of hypotheses noted

- Similarities and differences to other studies noted

- Limitations of study noted

- Avenues for future research provided

CONCLUSIONS

  • Were well written and concise

FORM, STYLE, AND SUBSTANCE

Text is clearly written, logically organized, length is appropriate

REFERENCES

- Relevant and comprehensive

Author Response

Thank you for your comments. 

Round 2

Reviewer 2 Report

Thank you. The authors have clarified my concerns. I would isuggest publication in current form.